# Factors influencing maternal death in Cambodia, Laos, Myanmar, and Vietnam countries: A systematic review

Pyae Phyo Win[ID]¹*, Thein Hlaing², Hla Hla Win³

1 Department of Public Health and Social Medicine, University of Medicine, Magwae, Myanmar, 2 District Public Health Department (Ministry of Health), Pyay District, Bago Region, Myanmar, 3 Department of Health and Social Sciences, STI Myanmar University, Yangon, Myanmar

* pph.win@gmail.com

## Abstract

### Background

A maternal mortality ratio is a sensitive indicator when comparing the overall maternal health between countries and its very high figure indicates the failure of maternal healthcare efforts. Cambodia, Laos, Myanmar, and Vietnam-CLMV countries are the low-income countries of the South-East Asia region where their maternal mortality ratios are disproportionately high. This systematic review aimed to summarize all possible factors influencing maternal mortality in CLMV countries.

### Methods

This systematic review applied "*The Preferred Reporting Items for Systematic Reviews and Meta-Analyses (PRISMA) Checklist (2020)*", Three key phrases: "Maternal Mortality and Health Outcome", "Maternal Healthcare Interventions" and "CLMV Countries" were used for the literature search. 75 full-text papers were systematically selected from three databases (PubMed, Google Scholar and Hinari). Two stages of data analysis were descriptive analysis of the general information of the included papers and qualitative analysis of key findings.

### Results

Poor family income, illiteracy, low education levels, living in poor households, and agricultural and unskilled manual job types of mothers contributed to insufficient antenatal care. Maternal factors like non-marital status and sex-associated work were highly associated with induced abortions while being rural women, ethnic minorities, poor maternal knowledge and attitudes, certain social and cultural beliefs and husbands' influences directly contributed to the limitations of maternal healthcare services. Maternal factors that made more contributions to poor maternal healthcare outcomes included lower quintiles of wealth index, maternal smoking and drinking behaviours, early and elderly age at marriage, over 35 years pregnancies, unfavourable birth history, gender-based violence experiences, multigravida and higher parity. Higher unmet needs and lower demands for maternal healthcare services

**Data Availability Statement:** All selected papers and their extracted information have not been published before and were saved in the document file of the principal reviewer. All important and applicable information were presented in this

report. A minimal data set (an excel file) was attached as the additional information. For unclear information and more data, the details of contact are +95-254555620 and pph.win@gmail.com.

**Funding:** The authors received no specific funding for this work.

**Competing interests:** The authors have declared that no competing interests exist.

occurred among women living far from healthcare facilities. Regarding the maternal healthcare workforce, the quality and number of healthcare providers, the development of healthcare infrastructures and human resource management policy appeared to be arguable. Concerning maternal healthcare service use, the provisions of mobile and outreach maternal healthcare services were inconvenient and limited.

## Conclusion

Low utilization rates were due to several supply-side constraints. The results will advance knowledge about maternal healthcare and mortality and provide a valuable summary to policymakers for developing policies and strategies promoting high-quality maternal healthcare.

## Introduction

Maternal death is a woman's death and it is directly associated with maternal causes of pregnancy, childbirth and puerperium between the 1st day of initiation of pregnancy and the 42nd day of termination of pregnancy,but does not concern other maternal death from accidental or incidental causes [1]. In 2020, for every 100,000 Live Births (LB), 152 mothers died globally and 147 mothers also passed away in South Asia Region due to preventable causes of maternal deaths such as disparities in geographical and socioeconomic status, marital, pregnancy and childbirth-related age, and contraception [2]. Many countries around the world calculate the maternal mortality ratio to determine their individual and social development. A maternal mortality ratio is a sensitive indicator when comparing the overall maternal health between the countries and its very high measure could be assumed as the failure of maternal healthcare efforts in a country. On the other hand, a low maternal mortality ratio indicates satisfaction with the overall status of women's health and the women's accessibility to reproductive healthcare services as well as reflects the responsiveness of the national health system to women's health needs [3].

For many years, the World Health Organization (WHO) has been exploring ways to overcome the inequalities in maternal mortality between low-income and high-income countries. In these ways, nutrition supplementation was provided andmaternal immunization was promoted through outreach strategies. Besides, training volunteer health workers (Axillary Midwives, Traditional Birth Attendants and Community Health Workers), and healthcare providers, upgrading maternal healthcare infrastructure and equipment, promoting accessibility to maternal healthcare services, and awareness were also included [3]. Moreover, in 2001, eight goals of the Millennium Development Goals (MDGs) were set and implemented to reduce the maternal mortality ratios [4], After adopting and implementing the WHO's maternal reduction strategies and global goals (MDGs-2001) among 189 countries, the global maternal mortality ratio (MMR) was reduced by 45%, MMR in Southeast Asia was reduced by 64%. Moreover, the global skill birth attendance rate was increased by 12%, global quality antenatal coverage was increased by 39%, and global contraceptive prevalence among married women was increased by 9% between 2000 and 2015. However, after analysis of MMR at the end of 2015, MMR was reduced by less than half of the MDG targeted settings and more effective interventions still need to reach the predetermined MDG goals among Low- and Middle-income Countries (LMICs) including CLMV countries [4]. In many LMICs, the responsibility

for changing the constant or slow declination of maternal mortality trends into the targeted line of the health-related Millennium Development Goals (MDGs) 5 remained a major challenge. In the goal of MDG 5, the first and second targets were "reducing the maternal mortality ratio by three-quarters" and "achieving universal access to reproductive healthcare services" at the end of the 25 years duration after 1990. Until 2017, many LMICs were under the level of the targeted achievement and there were nearly 800 pregnancy- and childbirth-associated deaths every day. More than 90% of global maternal deaths were contributed to LMICs [5].

Consequently, heads of states and governments from 189 countries adopted the 2030 Agenda launched by United Nations in September 2015 with the aims of ending all forms of poverty. The agenda included the 17 sustainable development goals (SDGs) and a goal was set to replace eight MDGs to end all preventable causes of maternal deaths. As announced by a United Nations Summit in the SDG context, countries have rallied along with a new goal to drive the action of reduction in maternal mortality by 2030. According to the agenda, global MMR is to be less than 70/100,000 LB and the MMR of each country is not to be more than twice the average of global MMR at the end of 2030. The CLMV countries are the low-income countries of the South-East Asia region where their maternal mortality ratios are disproportionately high [6]. They are also responsible to reduce their maternal mortality ratios to < 70/100,000 LB (Live Births) by 2030 according to the SDGs of the United Nations Summit. In taking action the stakeholders of the CLMV countries perceived that they lacked ownership, some SDG goals were less relevant to the local development contexts and the strategies were more likely to be related to donors' wants. Therefore, for the CLMV countries, their particular national strategic plans gave the preferences to their local needs and have been prioritized. Through the energetic work of the technical working group, more specific plans based on their local needs are sustainably developed, their achievements in maternal health are regularly reviewed and their developed plans, policies and achievements are assessed for the alignments of SDGs [6].

Maternal Death Surveillance and Response (MDSR) is recognized as another option for maternal death-reducing intervention as well as this concept is widely established globally as a sufficient intervention in the era of MDGs. It is a new approach in which causes of maternal deaths and the possible determinant factors are reviewed and the appropriate responses are undertaken according to the basis of the feedback and recommendation evolved from the MDSR establishment [7]. However, many established systems of MDSR from many countries including the CLMV countries frequently end with the first three stages of the action cycle: identifying and notifying all deaths, reviewing maternal deaths and analyzing and making a recommendation that does not reach the final stage: the action. As a result, similar maternal death events occur in future and MDSR becomes paperwork rather than the response element regarding maternal deaths. According to the survey on the MDSR implementation of the LMICs in 2015, the record showed that, of LMICs, only 86% have notified the maternal death cases and 85% reviewed determinants of the reported maternal death cases. 24% of LMICs had no MDSR review committee, and 54% of LMICs could not review MDSR at least biannually. This survey suggested that there were many gaps in MDSR policies and practices that need to be strengthened for eliminating preventable maternal deaths [7].

Historically, the CLMV countries have been explored, conquered, colonized and settled by many European countries, causing the CLMV countries less development for a long time. In evidence, Myanmar was part of British India from 1824 to 1948, and Cambodia Laos and Vietnam were part of French Indochina from 1887 to 1954 [8]. Consequently, in 2019, Gross Domestic Product per capita of CLMVs averaged 2,200 US$, still much lower than the ASEAN average of 12,979 US$ and the Asian average of 17,909 US$, showing less development [8]. In ASEAN, the CLMV countries have the highest maternal mortality rate in comparison with

other LMICs. The CLMV countries managed to reduce the maternal mortality rate by 32% to reach the target of a 2/3 reduction in MMR over 25 years (from 1990 to 2015) [2].

Particularly, as the average MMR of Cambodia from 1990 to 2020 was 276/100,000 LB, the measure in 2020 still indicates that 206/100,000 LB need to be reduced for achieving the SDG target by 2030 (<70/100,000 LB). For Lao PDR, its average MMR from 1990 to 2020 was 362/100,000 LB and its 2020 MMR data indicated that 292/100,000 LB need to be dropped between 2021 and 2030. In Myanmar, its average MMR from 1990 to 2020 was 186 and its 2020 MMR data pointed out that it needs to drop more than 116/100,000 LB to meet the SDG target (less than 70/100,000 LB). When looking at Vietnam, the average MMR from 1990 to 2020 was 199/100,000 LB. Also, Vietnam needs to remove more than 129/100,000 LB from the 2020 MMR figure to achieve the SDG target (less than 70/100,000 LB) [2]. The maternal mortality trends among CLMV countries (1990–2020) are shown in Fig 1 (Page number (7)).

Significant reductions in MMR were visible among the CLMV countries. When estimating the average annual MMR (per 100,000 LB) of reduction, 7.1 could be reduced in Cambodia, 14.9 in Laos, 0.4 in Myanmar and 3 in Vietnam. Actually, for achieving the SDG target of 70 maternal deaths (MD)/100,000 LB in 2030, Cambodia needs to reduce 20.6 MD/100,000 LB, Laos needs to reduce 36.2 MD/100,000 LB, Myanmar needs to drop 11.6 MD/100,000 LB and Vietnam need to reduce 12.9 MD/100,000 LB annually. In these cases, among CLMV countries, there are still huge differences between the average annual MMR of reduction and the annual MMR needed to achieve the SDG target [2]. Accordingly, the CLMV countries will be difficult in achieving the MMR-related SDG goals within the targeted period (by 2030). This systematic review aimed to identify socio-economic factors, maternal biological factors, maternal health awareness-associated factors, community-associated factors, maternal health need-associated factors, maternal healthcare workforce-associated factors, maternal healthcare service use-associated factors and maternal psychological factors influencing maternal deaths in CLMV countries.

## Material and method

### Research design

In this study, the researchers wanted to summarize all possible factors influencing the improvement of MMR in CLMV countries by limiting avoidable biases such as paper selection bias, data extraction bias, analysis and interpretation bias and confounding bias. Besides, the researchers intended to establish a better-quality review output through the critical appraisal of the methodological domains of the included studies. Because of these reasons, this study employed a design of the systematic review.

The researchers used "*The Preferred Reporting Items for Systematic reviews and Meta-Analyses (PRISMA) Checklist (2020)*", which includes 27 items in 7 sections. The research question of the review was "What are the factors influencing the improvement of maternal deaths in Cambodia, Laos, Myanmar, and Vietnam, CLMV countries?". To construct the question, the researcher applied the "PICOT" criteria (see Fig 2) (page 8). The appropriateness and specificity of the research question were checked with the FINER (Feasible, Interesting, Novel, Ethical, Relevant) criteria for being a good research question [9] (Table 1).

### Inclusion criteria

The criteria of the papers to be included in the review were totally based on the basic structure of the review question. The elements of PICOT (P = Population studies, I = Intervention, C = Comparison, O = Outcome and T = Time Frame) that were mentioned in Fig 2 (page.9) addressed the inclusion criteria of this review.

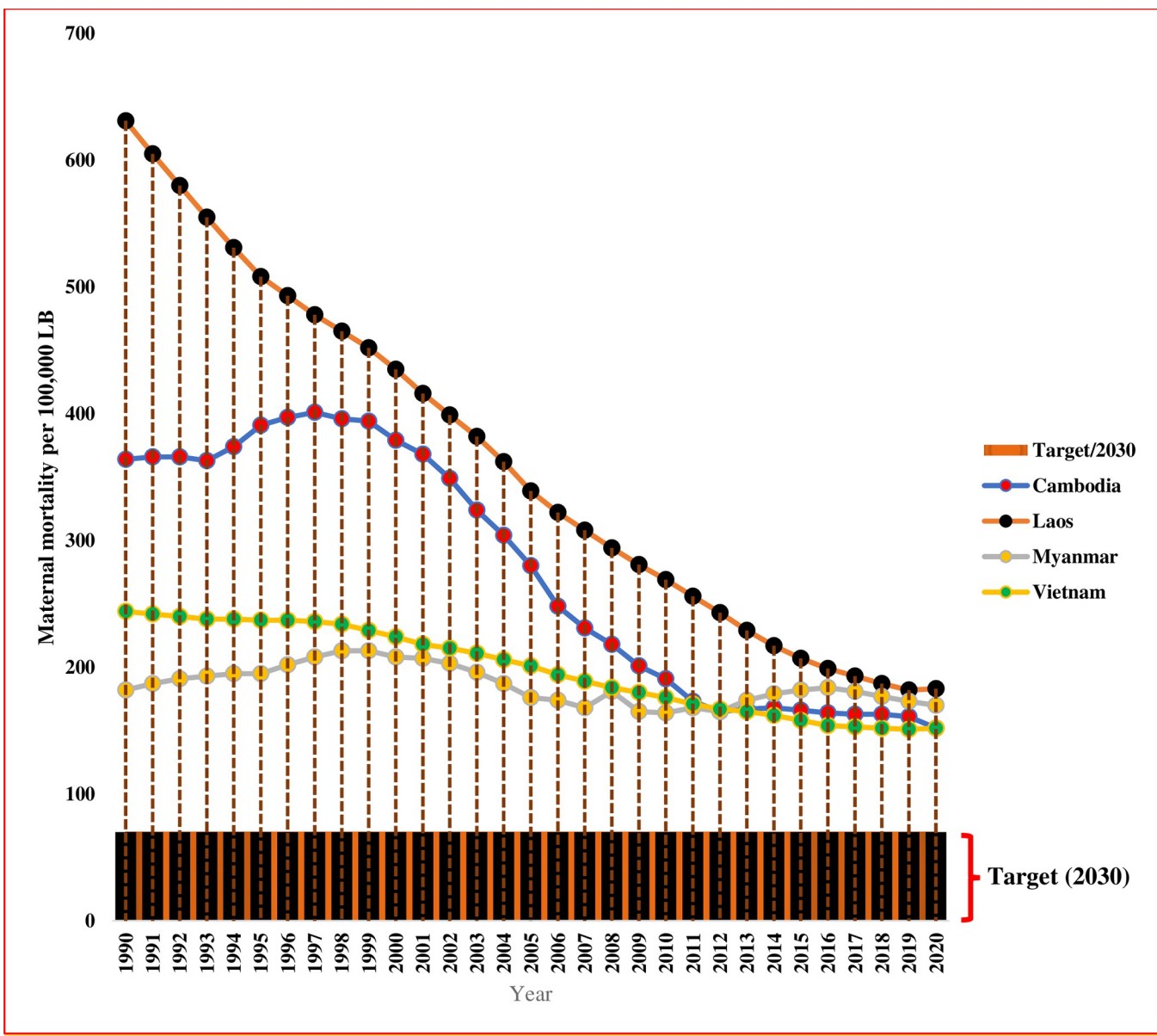

**Fig 1. Maternal mortality trends among CLMV countries (1990–2020).**

This systematic review typically focused on primary studies conducted using a specific type of qualitative and quantitative approach or a type of mixed qualitative and quantitative technique. The justification for the selection of only primary research was that the researcher wanted to collect more specific and conclusive information about the influencing factors of MMR. Besides, it was concluded that a methodology of mixed-methods study can also generate a convergence of more exact quantitative evidence and the behaviours, feelings, and attitudes of the targeted population [10]. Here, the researcher wanted to overcome the weaknesses of an approach with the strengths of another approach and vice versa in summarizing the evidence regarding the determinant factors of maternal death. This was a reason why the researcher primarily concentrated on a mixed-methods approach.

**P- Population**
Every married woman, Women aged 15-49 years, Volunteer health workers, Healthcare providers, Community members, Male partners, etc. from the CLMV countries

**I- Intervention**
The commitment to MDGs and SDGs, National health plans, Implementation of MDSR, Training, Maternal and reproductive healthcare services, Health education, Promotion of healthcare access, Upgrading Healthcare infrastructure and equipment, etc.

**O- Outcomes**
Socioeconomic factors, Maternal biological factors, Maternal health awareness-associated factors, Community-associated factors, Maternal health-needs-associated factors, Maternal healthcare workforce, Health system and service use-associated factors, etc.

**T- Time Frame**
All maternal death-related events and associated factors after 2010

**C- Comparison**
Not applicable

**PICOT**

**Fig 2. PICOT framework.**

**Table 1. Research question and FINER criteria.**

| FINER Criteria | Explanation |
|---|---|
| **Feasible (F)** | Adequate number of published papers with adequate sample sizes |
| | Presence of review team (Principal reviewer, Co-reviewer and Supervisor) |
| | Affordable in 3 months of study duration |
| | Exclude the purchased papers because this dissertation work was done by student's fund. |
| | Manageable in summarizing all possible influencing factors of MMR |
| **Interesting (I)** | Investigated the reasons for the very low improvement of MMR |
| | Explored different concepts of maternal deaths |
| | Support all complete and comprehensive information necessary for making the right decisions on maternal health- and mortality reduction-related policies and strategies |
| **Novel (N)** | Provided the complex issues of maternal deaths in a collective manner |
| | Provided updated and scientific evidence |
| **Ethical (E)** | Approved by the Ethics Committee of the School of Society Community and Health of the University of Bedfordshire, UK |
| **Relevant (R)** | No previous similar review |
| | Familiar with search engines and strategies |
| | Current |
| | Basic of future research |
| | Reflect on the clinical and public health policies |

The choice of studies, which were published between 2010 and 2021, inspired the researcher's interest in the subject. The 11-year evaluation period was not excessively long and could deliver essential, accurate, and appropriate data on the maternal mortality situation in CLMV countries. It was discussed that a long evaluation period of a systematic review might conclude the outdated issues that had already been addressed whereas a very short time bound of its evaluation might generate incomplete evidence [11]. The researchers wanted to explore the associated factors of MMR trends before and after the targeted date of MDGs (2015) as well as during the ongoing period of SDGs. The interesting point was that long-duration due to the global policy changes in 2015 from MDGs to SDGs. The MDGs were scheduled to conclude in 2015, and the argument over a post-2015 agenda is still ongoing.

This review was more likely to choose the studies conducted in four low-income countries of Asia: Cambodia, Laos, Myanmar, and Vietnam as the priority. The main rationale for selecting research that studied CLMV countries was that the researcher wanted to assemble all possible influencing factors on the slow declination of MMR and the reasons for being behind the global targeted settings within these countries.

## Exclusion criteria

This review excluded secondary analysis and evidence-based information that has previously been published as historical documents, systematic reviews or/and meta-analyses, reports, books, handouts, posters, factsheets, encyclopedias, student thesis, and conference proceedings. The reason for these exclusions was that the reviewers wanted to analyze the outcome information from the primary research efforts focusing on specific research question with detailed research methodology. The review also excluded others that did not agree with the inclusion criteria.

## Literature search

A strategic search for evidence was planned with five basic steps: development of the research question, selection of information sources, defining key search terms, search methods and evaluation of search results.

**Selection of information sources.** In this review, the researchers used only electronic resources, for which the Internet was essential. Firstly, the researchers listed online databases that are familiar and accessible to the principal researcher, co-researcher and research supervisor as well as relevant to the review title. The list included Hinari, Google Scholar, PubMed, Scopus, EMBASE, ScienceDirect, Web of Science, ERIC, IEEE Xplore, Directory of Open Access Journals (DOAJ), and JSTOR. Secondly, the research applied a simple random sampling procedure to choose three academic databases among lists of databases by limiting the database selection bias. Accordingly, Hinari, Google Scholar and PubMed were randomly selected as academic research databases that could provide valuable full-text articles with the permission of free download for the review analysis (Table 2).

**Defining key search terms.** To be the best linchpin between what the principal researcher and co-researcher want to search for and the key search terms inserted into the databases, the researchers used four key phrases: "Factors", "Maternal Mortality and Health Outcome", "Maternal Healthcare Interventions" and "CLMV Countries". To be a comprehensive search, the researchers expanded these four key phrases by using further similar words. To be provided with more precise search results, the researcher applied two Boolean Operators: "OR" (Narrows Search), "AND" (Broadens Search) and one search modifier "Double Quotation Mark" (Table 3).

**Table 2. List of the selected academic research databases and their descriptions.**

| Names of Databases | Available At | Description |
|---|---|---|
| Scopus | https://www.scopus.com/home.uri | Provided by Elsevier<br>Covered the approximately 71 million items<br>1.4 billion references<br>Multi disciplines<br>Free and full access |
| Web of Science | https://access.clarivate.com/login?app=wos&alternative=true&shibShireURL=https:%2F%2Fwww.webofknowledge.com%2F%3Fauth%3DShibboleth&shibReturnURL=https:%2F%2Fwww.webofknowledge.com%2F%3Fmode%3DNextgen%26action%3Dtransfer%26path%3D%252Fwos%252Fwoscc%252Fbasic-search%26DestApp%3DUA&referrer=mode%3DNextgen%26path%3D%252Fwos%252Fwoscc%252Fbasic-search%26DestApp%3DUA%26action%3Dtransfer&roaming=true | Provided by Clarivate<br>Covered approximately 110 million items<br>1.4 billion references<br>Access by institutional subscription only (Free and full access) |
| PubMed | https://pubmed.ncbi.nlm.nih.gov/ | Provided by NIH<br>Covered approximately 30 million items<br>Medicine and Biological Science<br>Free and full access |
| ERIC | https://eric.ed.gov/?multimedia-searching | Provided by US (Department of Education)<br>Covered approximately 1.3 million items<br>Education Science<br>Free and full access |
| IEEE Xplore | https://ieeexplore.ieee.org/Xplore/home.jsp | Provided by the Institute of Electrical and Electronics Engineers<br>Covered approximately 5 million items<br>Engineering<br>Free and full access |
| ScienceDirect | https://www.sciencedirect.com/ | Provided by Elsevier<br>Covered the approximately 16 million items<br>Multi disciplines<br>Free and full access |
| DOAJ | https://doaj.org/ | Provided by DOAJ<br>Covered approximately 4.3 million items<br>Multi disciplines<br>Free and full access |
| JSTOR | https://www.jstor.org/ | Provided by ITHAKA<br>Covered approximately 12 million items<br>Multi disciplines<br>Free and full access |
| Hinari | https://partnership.who.int/hinari | Provided by WHO<br>Covered up to 21000 journals, up to 69,000 e-books, and up to 115 other information sources<br>Multidiscipline<br>Free and full access |
| Google Scholar | https://scholar.google.com/ | Provided by Google<br>Covered approximately 100 million items<br>Multidiscipline<br>Free and full access |
| EMBASE | https://www.embase.com/ | Provided by Elsevier<br>Covered more than 32 million items<br>Multidiscipline<br>Free and full access |

**Search method.** *Organizing review team.* In this systematic review, a review team was organized with the principal researcher, a co-researcher (an experienced reviewer, an MSc Public Health holder, had many publications of systematic review) and a supervisor who is a professor and MSc Public Health Course coordinator of Science & Technology International Myanmar University (STIMU) to be transparent and better practice in the systematic review

**Table 3. Application of key search terms.**

| Main Search Phrases | Similar Terms | Usage of Boolean Terms and Search Modifiers |
|---|---|---|
| Factors | Obstacles, Constraints, Problems, Barriers, Determinants, Causal factors Associated Factors Significant Factors Threats Challenges | "Influencing factors" OR "Obstacles" OR "Constraints" OR "Problems" OR "Barriers" OR "Determinants" OR "Casual factors" OR "Associated Factors" OR "Significant Factors" OR "Threats" OR "Challenges" |
| Maternal Mortality and Health Outcome | Maternal mortality Maternal mortality rate Maternal mortality ratio MMR Pregnancy-related death Childbirth-related death Puerperium-related death Antenatal death Postnatal death Maternal health outcomes | "Influencing factors" OR "Obstacles" OR "Constraints" OR "Problems" OR "Barriers" OR "Determinants" OR "Casual factors" OR "Associated Factors" OR "Significant Factors" OR "Threats" OR "Challenges" AND "Maternal deaths" OR "Maternal mortality" OR "Maternal mortality rate" OR "Maternal mortality ratio" OR "MMR" OR "Pregnancy-related death" OR "Childbirth-related death" OR "Puerperium-related death" OR "Antenatal death" OR "Postnatal death" OR "Maternal health outcomes" |
| Maternal Healthcare Intervention | Antenatal care Labour care Delivery care Postnatal care Maternal healthcare strategies Maternal healthcare services Family planning Contraceptives Maternal health promotion Maternal health education Maternal healthcare facility Maternal healthcare infrastructure Maternal healthcare equipment MDSR MDGs SDGs Prevention of mother-to-child transmission PMCT HIV counselling and testing Financial support Training Maternal nutrition | "Influencing factors" OR "Obstacles" OR "Constraints" OR "Problems" OR "Barriers" OR "Determinants" OR "Casual factors" OR "Associated Factors" OR "Significant Factors" OR "Threats" OR "Challenges" AND "Maternal deaths" OR "Maternal mortality" OR "Maternal mortality rate" OR "Maternal mortality ratio" OR "MMR" OR "Pregnancy-related death" OR "Childbirth-related death" OR "Puerperium-related death" OR "Antenatal death" OR "Postnatal death" OR "Maternal health outcomes" AND "Maternal healthcare intervention" OR "Antenatal care Labour care" OR "Delivery care Postnatal care" OR "Maternal healthcare strategies" OR "Maternal healthcare services" OR "Family planning" OR "Contraceptives" OR "Maternal health promotion" OR "Maternal health education" OR "Maternal healthcare facility" OR "Maternal healthcare infrastructure" OR "Maternal healthcare equipment" OR "Maternal death surveillance and response" OR "MDSR" OR "MDGs" OR "SDGs" OR "Prevention of mother to child transmission" OR "PMCT" OR "HIV counselling and testing" OR "Financial support" OR "Training" OR "Maternal nutrition" |
| CLMV Countries | Cambodia Laos Lao People's Democratic Republic Lao PDR Myanmar Vietnam | "Influencing factors" OR "Obstacles" OR "Constraints" OR "Problems" OR "Barriers" OR "Determinants" OR "Casual factors" OR "Associated Factors" OR "Significant Factors" OR "Threats" OR "Challenges" AND "Maternal deaths" OR "Maternal mortality" OR "Maternal mortality rate" OR "Maternal mortality ratio" OR "MMR" OR "Pregnancy-related death" OR "Childbirth-related death" OR "Puerperium-related death" OR "Antenatal death" OR "Postnatal death" OR "Maternal health outcomes" AND "Maternal healthcare intervention" OR "Antenatal care" OR "Labour care" OR "Delivery care Postnatal care" OR "Maternal healthcare strategies" OR "Maternal healthcare services" OR "Family planning" OR "Contraceptives" OR "Maternal health promotion" OR "Maternal health education" OR "Maternal healthcare facility" OR "Maternal healthcare infrastructure" OR "Maternal healthcare equipment" OR "Maternal death surveillance and response" OR "MDSR" OR "MDGs" OR "SDGs" OR "Prevention of mother to child transmission" OR "PMCT" OR "HIV counselling and testing" OR "Financial support" OR "Training" OR "Maternal nutrition" AND "CLMV countries" OR/AND "Cambodia" OR/AND "Laos" OR "Lao People's Democratic Republic" OR "Lao PDR" OR/AND "Myanmar" OR/AND "Vietnam" |

**Table 4. Contributions of review team members to review processes.**

| Review Process | Contributions to review processes | | |
|---|---|---|---|
| | Principal Reviewer (PR) | Co-reviewer (CR) | Supervisor (S) |
| Review Question | PR | | |
| Research Design | PR | | S |
| Research Protocol | PR | CR | S |
| Literature Search | PR | CR | |
| Study Selection | PR | CR | |
| Critical Appraisal | PR | CR | |
| Data Extraction | PR | CR | |
| Data Synthesis and Interpretation | PR | | |
| Paper Writing | PR | | |
| Reviewing and editing | | CR | S |
| Rewriting | PR | | |
| Submitting Paper | PR | | |

through a complete dual review process and increase the number of relevant studies and quality of the review as well as to reduce unnecessary biases (Table 4). The independent literature search was completed in Hinari, Google Scholar and PubMed from 13 July 2022 to 1st September 2022.

*Developing tools.* The researchers developed, pretested and applied three tools useful for data search, paper selection and data extraction.

**Excel sheet for search terms and search results.** arranged all core search terms to be used by both principal researcher and co-researcher specifically and equally. The number of titles and abstracts screened via databases was recorded in the sheet according to the search result of each search.

**Inter-researcher Agreement (IRA) form.** used for carrying out the proper assessment of the includable papers by each researcher, providing reasons for exclusions and taking agreement between researchers. The IRA form included the variables: paper title, authors' names, paper code, search engine, agreements of the researcher ("Yes", "No", and "Maybe") and reasons for exclusion. If there were disagreements between researchers, re-screening practices on "No" and "Maybe" comments were ensured. When developing some conflicts between two researchers, the agreement was based on the decisions of the research supervisor.

**Data Extraction Template (DET).** was developed for collecting the required information and pre-tested on a set of 10 selected papers for its clearness, understandability, applicability and representativeness of all included studies. DET was structured with 21 items in three sections. The first section included "Paper title", "Author(s)", "Publication year", "Country of study", "Study objective(s)", "Study design(s), "Sampling method(s)", "Study population", "Data collection tool(s)", "Sample size", "Data analysis tools", the second section included "Socioeconomic factors", "Maternal biological factors", "Maternal health awareness-associated factors", "Community-associated factors", "Maternal health needs-associated factors", "Status and quality of maternal healthcare workforce", and "Health system- and maternal healthcare service use-associated factors and the third section included "Quality scores" and "Study limitations".

**Evaluation of search results.** The researchers stopped a search when a number of search results was not very different and the appeared papers were frequently duplicated or when the relevant papers did not find after screening three pages of search results. Importantly, the researchers screened the downloaded papers frequently whether the abstract contents of the papers met the predefined review objectives for determining a need for further search.

## Study selection

Through three main stages (identification stage, screening stage and selection stage) of the PRISMA study selection flow diagram, the relevant ones were completely selected. The selection process of the review was a double-screening approach and two researchers consistently applied the predetermined inclusion criteria to avoid systematic errors. Details of selection processes were explained in the finding section (page 22).

## Information extraction

Information extraction of this review was completed by the independent works of two data extractors (researchers), for which a self-developed data extraction template (DET) with a manual guide was used to collect the most applicable variables presented in 75 published papers which met with the review objectives. The supervisor checked the reliabilities, completeness, and correctness of the information extraction processes and the extracted information whenever each data extractor completed a set of 10 selected papers. The DETs were exchanged between two data extractors for taking the agreements on information extractions.

## Quality appraisal

A work of critical appraisal in a systematic review is incorporated to assess the researchers' abilities to control the possibilities of bias in their research materials, methods and analysis. It was also exercised to reduce irrelevant content, focus on the relevant content and provide the review output with high-quality evidence [12]. However, the researcher practiced the appraisal work for improving academic knowledge and skill only and the practice of the researcher and decisions of the papers' quality statuses did not concern with inclusion/exclusion of the papers. In this work, JBI critical appraisal tool sets were chosen and applied according to the study designs of the included papers. JBI stands for Joanna Briggs Institute and its appraisal tool sets are available at: https://jbi.global/critical-appraisal-tools. The researchers' reason for JBI use was that the appraisal tool set was capable of assessing the methodological qualities of the different study designs and trustworthiness and relevancy of research outputs. Here, a checklist for analytical cross-sectional study, checklist for case-control study, checklist for the cohort study, checklist for prevalence study, checklist for qualitative study and checklist for the experimental study were applied.

## Data analysis

The data analysis of the review included two stages: descriptive analysis of the general information of the included papers and qualitative analysis of key findings extracted from the included papers. For descriptive analysis, excel 2019 was applied and frequencies, percentages, tables and graphs were calculated and illustrated to show the constituents of background characteristics of the included papers. In qualitative analysis, among different approaches: content analysis, narrative analysis, discourse analysis, framework analysis and grounded theory, this review analyzed all key extracted information through content analysis. The reason for applying content analysis, the analysis help researcher to identify themes, patterns and relationships of the key extracted information, apply a manual method to exercise the open and axial coding of similar findings and categorize eight themes (socio-economic factors, maternal biological factor, maternal health awareness-associated factor, maternal health need-associated factor, maternal psychological factor, status and quality of the maternal healthcare workforce, maternal healthcare service use-associated factor and community-associated factor) as predefined in the review objective setting. Finally, the highlighted major themes were summarized and

assembled according to the representatives of eight themes: socio-economic factors, maternal biological factors, maternal health awareness-associated factors, community-associated factors, maternal health need-associated factors, maternal healthcare workforce-associated factors, maternal healthcare service use-associated factors and maternal psychological factors.

## Ethical consideration

This systematic review ha vsery few ethical concerns because the researchers did not directly collect any personal, sensitive and confidential information of human subjects [13]. Besides, before commencing this review, the researchers submitted the review protocol with an ethical form to both institutional review boards of Science & Technology International Myanmar University (STIMU), Myanmar and the University of Bedfordshire, the UK for approval. Both boards approved the protocol on 13th July 2022 with the Ethics Committee of the School Society Community and Health (Public Health) (SSCHREC) application number (STI011-6-2022-3-003). During the review, the researcher used only publicly accessible documents and abstained from abnormal minimization or maximization of the original research findings.

## Results

### Summary of paper selection process

Two researchers first identified 1112 titles and abstracts by searching PubMed (n = 348), Google Scholar (n = 496), and Hinari (n = 268) records. Following the completion of the identification, 378 records were retained for further screening, while 734 records were excluded due to duplications and ineligibilities. And the 378 full-text papers were screened using the inter-reviewer agreement form. Two reviewers independently checked and counter-checked 174 papers against the review's pre-determined inclusion/exclusion criteria. The reviewers rejected 99 of the 174 papers for a variety of reasons. Finally, 75 (6.7% of total identified records) full-text papers remained for qualitative analysis (see Fig 3) (page. 23).

### Background information of included studies

Since this study concentrated on the CLMV countries, all included studies took place in these four countries. 27 of the 75 studies were from Myanmar, 23 from Laos, 13 from Cambodia, and 12 from Vietnam.

Among the 75 studies, 54 discussed maternal death-related socioeconomic factors, 42 discussed maternal health needs, 30 discussed the influencing factors of maternal health literacy promotion, 24 discussed the status and quality of maternal healthcare workforce, and 23 discussed health system and maternal healthcare services. 13 disclosed maternal death-related maternal biological factors, while the remaining 19 disclosed community-associated and maternal psychological factors influencing maternal health.

In terms of research designs, 36 studies were cross-sectional studies, 18 were qualitative research designs, 11 were secondary data analysis, 7 were cohort research designs, 2 were case-control research designs, and 1 was survey methods.

In this review, 19 studies were published between 2010 and 2014, and 25 between 2015 and 2018 and 31 between 2019 and 2022. And the total sample size for this systematic review was 2,361,463, with a sample size ranging from 23 to 2,214,339.

The quantitative information in this review was gathered using semi-structured interview questionnaires (n = 62) and the qualitative information was gathered through focus group discussions and key informant interviews (n = 13).

**Identification of studies via three databases**

| Identification | Records identified from:<br>PubMed (n = 348)<br>Google Scholar (n = 496)<br>Hinari (n = 268) | → | Records removed *before screening*:<br>Duplicate records removed (n = 478)<br>Records marked as ineligible by automation tools (n = 0)<br>Records removed for other reasons (n = 256) = total 734 |
|---|---|---|---|

| Screening | Records screened<br>(n = 378) | → | Records excluded<br>(n = 204) |
|---|---|---|---|
| | Reports sought for retrieval<br>(n= 174) | → | Reports not retrieved<br>(n = 0) |
| | Reports assessed for eligibility<br>(n = 174) | → | Reports excluded:<br>Reports (n=17)<br>Quitting and associated factors (n=11)<br>Incomplete papers (n=16)<br>Review (n=9)<br>Not English story (n=15)<br>Systematic review (n=6)<br>Ineligible time (n=13)<br>Duplicated paper (n=7)<br>Not CLMV countries (n=5)<br>Total = 99 |

| Included | Studies included in review<br>(n = 75) |
|---|---|

**Fig 3. Flow diagram of paper selection process.**

In terms of data analysis tools, 32 used multivariate analysis, 20 used descriptive analysis, 13 used thematic coding analysis, and the remaining 10 used logistic regression, operational analysis, sequential regression, and the student t-test.

According to the JBI criteria, some cross-sectional studies (12.5%, n = 6) did not mention pre-testing data collection instruments and Cronbach alpha values as well as the standardization of the instruments. And (25%, n = 12) studies also did not explain how they identified the confounding factors. However, all of (100%, n = 18) qualitative studies, all of (100%, n = 2) case-control studies and 6 out of 7 (85.7%) cohort studies were found to be fully completed with JBI criteria (Table 5).

**Table 5. Appraisal results per JBI checklist.**

**Cross-sectional Studies (n = 48)**

| Criteria | Number of the included papers | | | |
|---|---|---|---|---|
| | Yes | No | Unclear | Not Applicable |
| Defining inclusion criteria | 48 | | | |
| Details of study population and setting | 48 | | | |
| Valid and reliable measurement of exposure | 48 | | | |
| Use of standardized measurement | 42 | | 6 | |
| Identification of confounding factors | 36 | 12 | | |
| Dealing with confounding factors | 35 | 13 | | |
| Valid and reliable measurement of outcome | 48 | | | |
| Use of appropriate statistical analysis | 48 | | | |
| Qualitative Studies (n = 18) | | | | |
| Congruity between stated philosophical perspective and methodology | 18 | | | |
| Congruity between the research methodology and the research question or objectives | 18 | | | |
| Congruity between the research methodology and data collection methods | 18 | | | |
| Congruity between the research methodology and data analysis | 18 | | | |
| Congruity between the research methodology and the interpretation of results | 18 | | | |
| Statement locating the researcher culturally or theoretically | 18 | | | |
| Influence of the researcher on the research | 18 | | | |
| Adequate representativeness of participants, and their voices | 18 | | | |
| Ethical consideration | 18 | | | |
| The link between conclusions and the analysis and interpretation | 18 | | | |
| Cohort Studies (n = 7) | | | | |
| Recruitment of two groups from the same population | 7 | | | |
| Measurement of exposures in both groups similarly | 7 | | | |
| Valid and reliable measurement of exposure | 7 | | | |
| Identification of confounding factors | 7 | | | |
| Dealing with confounding factors | 6 | 1 | | |
| Absence of outcomes in both groups before the study | 7 | | | |
| Valid and reliable measurement of outcome | 7 | | | |
| Sufficient follow-up time | 7 | | | |
| Complete follow-up time, if no, reason | | | | 7 |
| Addressing incomplete follow-up time | | | | 7 |
| Use of appropriate statistical analysis | 7 | | | |
| Case-control Studies (n = 2) | | | | |
| Comparable groups | 2 | | | |
| Appropriate match of case and control | 2 | | | |
| Use of same criteria | 2 | | | |
| Use of standardized measurement | 2 | | | |
| Use of equal measurement between case and control | 2 | | | |
| Identification of confounding factors | 2 | | | |
| Dealing with confounding factors | | | | 2 |
| Valid and reliable measurement of outcome between case and control | 2 | | | |
| Enough exposure time | 2 | | | |
| Use of appropriate statistical analysis | 2 | | | |

### Factors influencing maternal death

**Socio-economic factors.** Women with lower socioeconomic status had less access to quality maternal and reproductive healthcare services and were at a higher risk of maternal death (Table 6). According to the findings of this review, family income was a significant factor that was strongly associated with maternal and reproductive healthcare access and maternal death outcomes [14, 15]. It was also found that poorer women were about 5 times (95% CI, 2.2–9.6) more likely than richer women to object to high-cost antenatal care services, obstetric and gynecological care, and ultrasound machine management [16–22].

Furthermore, occupation of both women and husbands was important for positive maternal health outcomes. Pregnancies were 40–45% less likely to be delivered in a hospital if their husbands worked in agriculture or unskilled manual labor [23] and the chances of developing post-partum complications were about 30% if the pregnancies were manual or agricultural laborers, The repetitive abortion rates (30–40%) among female sex workers were 20–26% higher compared to these rates (10–14%) among farmers [24].

It was also discovered that maternal health care services utilization and poor maternal health outcomes were highly correlated with mother education [25–27]. Furthermore, a study found that pregnant women with higher education received more than 7 times of antenatal care, while women with middle education levels received approximately 6 times of antenatal care, women with primary education levels received approximately 5 times of antenatal care, and those with no education received less than 4 times of antenatal care (P0.000) [28].

In terms of male involvement, having a supportive husband was positively correlated with current contraceptive use, antenatal care practices, and delivery plans. Women who had husbands who supported contraception were more likely to succeed in family planning than other women [29]. Similarly, the studies concluded that favorable decisions made by husbands, gender, and societal norms had a significant impact on women's access to antenatal care, contraceptive choice, and delivery options [30, 31]. However, it was discovered that most husbands' perspectives were that discussions about pregnancy care, family planning, and delivery cases were not their concerns [32].

In terms of residence, rural women (54.6%) were more likely than urban women (61.7%) to have lower institutional delivery and skilled birth attendance, indicating that geographical barriers influenced the availability of maternal healthcare services and maternal health outcomes (P < 0.01) [23, 32]. Long geographical distances and difficult-to-reach areas were major factors in the second delay of maternal deaths, and it was determined that the second delay was responsible for more than 80% of rural mother deaths [26].

Furthermore, rather than financial constraints, the delay in seeking care or the first delay that could lead to the mother's death was heavily influenced by certain social and cultural beliefs. Pregnant women's lack of empowerment, the influence of spiritual healers, a close relationship with traditional birth attendance, and medication misuse were all important factors in the delay in seeking care [18, 29, 33–35]. It was also discovered a culture in some ethnic groups that may lead to maternal death, in which pregnant women had to go into a forest for their births, where births were performed by themselves or with the assistance of their husbands, and then spend three days afterwards [21].

**Maternal biological factors.** Early and older marriage ages were found to increase the likelihood of not receiving quality antenatal care, and elderly pregnant women were more likely to rely on traditional therapies. Early childbirth age was statistically associated with birth-related complications and missed postnatal follow-up care (Table 7) [22, 34].

Furthermore, higher parity was significantly associated with gestational diabetes mellitus (GDM), which in turn was significantly associated with maternal mortality [46]. It was argued

**Table 6. The influence of socioeconomic factors on accessing maternal health care services and high risk of maternal deaths.**

| Exposures and Studies | Outcomes | Strength of Difference |
|---|---|---|
| **Family income** | | |
| Ir et al., 2010 [15]<br>Ye et al., 2010 [16] | In comparison to women with low family incomes, women with high incomes were more likely to accept quality antenatal care services | OR = 3 (1.2–5.7) |
| Binder-Finnema et al., 2015 [17]<br>Dingle, Powell-Jackson and Goodman, 2013 [18]<br>Do Kim Ngan et al., 2013 [19]<br>Sanaphay et al., 2014 [20] Sychareun et al., 2012 [21]<br>Sychareun et al., 2013 [22]<br>Tran et al., 2019 [23] | Overall antenatal care insufficiencies were more likely to be found among rural and poor pregnant women than in their counterparts | OR = 5 (2.0–10.4) |
| Douangphachanh et al., 2012 [37]<br>Kawaguchi et al., 2021 [33] Maung et al., 2020 [31] Milkowska-Shibata et al., 2020 [38] | A strong significant association occurred between high household income and delivery at health facilities | OR = 2.4<br>P<0.000 |
| Goland, Hoa and Målqvist, 2012 [39] | Among ethnic minority women, the poor had a higher chance of not receiving professional antenatal and obstetric care and a higher chance of not getting competent birth attendance | OR = 6.27 (2.37–16.6)<br>OR = 25.5 (, 11.4–56.8) |
| Thandar et al., 2019 [40] | Women in the second, middle and fourth quintiles of the wealth index showed better maternal health outcomes than those in the lowest quartile | P<0.007<br>P<0.08<br>P<0.019 |
| Khaing et al. 2019 [41] | A high level of family income was a significant predictor of seeking sufficient frequencies of antenatal, delivery and postnatal care as well as positive maternal health outcomes | AOR = 1.95 (0.99–3.84) |
| **Occupation** | | |
| Khine and Sawangdee, 2022[42] | Pregnant mothers who were general officers and managers were more likely to receive qualified antenatal care | OR = 1.06–1.1 |
| Sakuma et al., 2019 [43] | Being farmers were the determinants of low outcomes of maternal and child healthcare services. | P<0.002 |
| Duff et al., 2018 [44] | Being female sex workers had more chance to become physical/sexual violence and high risk for unwanted pregnancy. | AOR = 1.75 (1.10–2.78)<br>P<0.003 |
| Sopheab et al., 2015 [45] | Being workers in karaoke bars was more likely to be at risk of induced abortion and its consequences than other employees. | OR = 3 (2.7–8.36) |
| **Education** | | |
| Douangphachanh et al.,2012 [37]<br>Phathammavong et al., 2010 [36]<br>Tran et al., 2019 [23]<br>Ye et al., 2010 [16] | Educated mothers and their partners had<br>• higher likelihood of having gone to the ANC<br>• lesser probability of home births<br>• better figures of maternal health outcomes | • OR 5.3 (P<0.000)<br>• OR 5.1, (P<0.000)<br>• OR 4.1, P<0.000) |
| Goland, Hoa and Målqvist, 2012 [39] | Illiterate women had<br>• higher probability of not receiving the recommended antenatal and postnatal care<br>• giving birth with unskilled birth attendance<br>• suffering from worse pregnancy-related complications | • OR = 2.95 (1.70–5.12)<br>• OR = 6.33 (3.33–12.0)<br>• (P<0.033) |
| Yasuoka et al. 2018 [46] | Women with no education were five times less likely to reach the recommended four times of quality antenatal care | AOR = 5.5 (1.74–17.37) |
| **Male involvement** | | |
| Sychareun et al., 2012 [21] | The delivery process, maternal and child nutrition and contraceptive use of rural women were significantly influenced by their husbands and family members | P<0.003 |
| Alvesson et al., 2012 [34] | Male involvement was important for motivating, psychological support and financial support to be safe, happy and healthy mothers' lives and was highly associated with positive maternal health outcomes | AOR-6.08–1.48–25.97, (P<0.012) |
| **Residence & Ethnicity** | | |
| Milkowska-Shibata et al., 2020 [38] | Rural women were more likely to miss follow-up antenatal care and the required five antenatal appointments. | OR = 2.92 (1.58–5.38) (P<0.01) |
| Thandar et al., 2019 [40] | Rural geographic location was related to the indirect causes of maternal deaths such as low maternal health knowledge, higher prevalence of maternal smoking and drinking and traditional cultures affecting maternal health | |

*(Continued)*

**Table 6.** (Continued)

| Exposures and Studies | Outcomes | Strength of Difference |
|---|---|---|
| Wai *et al.*, 2016 [32] | Rural geographic location was related to the direct causes of maternal deaths such as<br>• low antenatal care coverage and<br>• higher delivery by traditional birth attendance | • OR = 0.36 (0.29–0.43)•<br>OR-0.29 (0.24–0.35) |
| Phathammavong *et al.*, 2010 [36] | Compared to highland mothers, the mothers from low and middle land were more likely to receive full times of antenatal care.Compared to mothers from the villages with rural health centres, mothers from the villages without rural health centres are<br>•less likely to have full-time antenatal care<br>• more likely to give their births at home | • OR = 4–7<br>• OR = 3.5 (1.95–6.35)<br>• OR- = 4.27 (1.81–10.09) |
| Goland, Hoa and Målqvist, 2012 [39] | Pregnant women living in a poor household had risk of not getting quality antenatal care as compared to those living in a non-poor household. | OR = 9.69 (5.15–18.24) |
| **Health-seeking practice & culture** | | |
| Samandari, Speizer and O'Connell, 2010 [30] | Lower probabilities of using contraception and higher probabilities of induced abortion were due to the ultimate decision-making power of the spouses and no power of women for any decision | |
| Sychareun *et al.*, 2012 [21] | During birth at home, traditional healers, the impact of mother-in-law and the influences of family members and neighbours were other problematic issues leading to delay in seeking care | |
| Sychareun *et al.*, 2013 [22] | Respecting the elders from the communities and abnormal beliefs in ghosts and spirits, calling a religious ritual before seeking care were found to be major contributors to delays | |

that the majority of maternal deaths occurred between the ages of 30 and 39, and that the majority of maternal deaths from eclampsia and postpartum hemorrhage occurred in this age range [26]. It was also determined that unmarried teenagers and young mothers had higher rates of preterm birth, unsafe abortion, and postpartum hemorrhage [47]. On the other hand, it was discovered that pregnancies over 35 years old with maternal underweight had a higher chance of abortion (67–85%), preterm delivery (13–30%), and low birth weight (66–67%) [22].

**Maternal health awareness-associated factors.** Mothers who were unaware of pregnancy-related signs and symptoms for referral, the benefits of ANC services, and the consequences of unplanned pregnancy had a higher risk of poor maternal health outcomes (Table 7). Similarly, women with low maternal health awareness were more likely to have a home birth with traditional birth attendance and a low prevalence of quality antenatal care coverage [28, 34]. According to one study, approximately one-third of HIV-positive pregnant women had less access to maternal health education and family planning education [48].

Furthermore, young women and primigravidae had significantly lower rates of sexual reproductive health discussion [47] and rural mothers had a lower awareness of pregnancy and childbirth complications, as well as postpartum dietary restrictions [49, 50].

**Maternal health need-associated factors.** Access to healthcare facilities was a significant determinant of high maternal mortality (Table 8). Due to the examination of unskilled axillary midwives, the absence of appropriate laboratory equipment, and the lack of examination instruments, women from remote areas frequently missed their pregnancy-induced hypertensive conditions [34]. It was also reported that the majority of rural mother deaths were caused by a lack of access to skilled birth attendance and unskilled and incorrect management of traditional birth attendance, as well as the consequences of birth-related complications being more prevalent among rural pregnant women managed by unskilled birth attendance [22, 36, 45]. Contrary to popular belief, wealthy pregnancies had higher demand for HIV testing, Syphilis testing, ultrasound screening, skilled birth attendance, high-quality antenatal care services, and institutional deliveries [17, 33].

**Table 7. The influence of maternal factors on accessing maternal health care services and high risk of maternal deaths.**

| Exposures and Studies | Outcomes | Strength of Difference |
|---|---|---|
| **Maternal biological factors** | | |
| Douangphachanh *et al.*, 2012 [37] | Compared to women with more than two living children, those with less than two children were more likely to take safe and institutional deliveries | (OR-2.3, P<0.001) |
| Målqvist *et al.*, 2013 [26] | Older age at pregnancy and children and multiporous women were highly associated with maternal deaths, However, maternal age was not linked to pregnancy care and birth location | (P<0.000) (P = 0.678). |
| Gilder *et al.*, 2014 [47] | Pregnancy age more than 35 years had higher GDM prevalence than pregnancy age less than 35 years. | 10.1% (95%CI- 6.1–14.0) versus 5.3% (95%CI- 2.3–8.2) |
| Thandar *et al.*, 2019 [40] | Old-age pregnancies and pregnancies who had a family history of diabetes were at high risk of GDM, GDM was also associated with other underlying diseases linked to maternal deaths | (P<0.008) |
| Khaing *et al.*, 2019 [41] | Women who had a history of childbirth with skilled birth attendance, low parity and were 25–34 years had double the likelihood of giving birth with skilled birth attendance compared to their counterparts. | AOR = 2.47; (1.15–5.30) |
| Nguyen *et al.*, 2018 [52] | Pregnant women aged between 25–35 years were significantly associated with<br>• low fruit intake and<br>• gestational weight gain | (P<0.05) (P<0.02). |
| **Maternal health awareness-associated factors** | | |
| Ye *et al.*, 2010 [16] | Women who were highly informed about maternal and reproductive healthcare and related services (e.g., HIV testing) were more likely to receive ANC services than their counterparts. Better maternal health outcomes were highly associated with the maternal positive attitudes regarding ANC services | OR = 6.5(2.4–17.6) OR = 3 (1.3–7.1) |
| Khine and Sawangdee, 2022 [42] | Pregnancies who were aware of the danger signs of pregnancy and childbirth were more likely to take institutional deliveries and postpartum care. | OR = 3 |
| Yanagisawa *et al.*, 2015 [53] | Women who received antenatal care from skilled healthcare providers were<br>• more likely to receive HIV and nutritional counselling and<br>• more likely to be high HIV testing and low anaemia | OR = 6–10 OR = 2.499 (1.746–3.578) |
| Kikuchi *et al.*, 2018 [54] | The study by had a significant association between high awareness levels of postnatal care and continuation of postnatal care | (P<0.000). |
| Khaing *et al.*, 2019 [41] | Women who had previous pregnancy and newborn care experiences were more likely to accept delivery and postnatal care provided by skilled birth attendants. | AOR-13.53 (3.13–58.40) |
| Horiuchi *et al.*, 2020 [55] | Women who had no prior maternal health literacy promotion had a lower chance of getting childbirth and postnatal care by skilled birth attendance than those who had maternal health literacy promotion before pregnancy. | 79% versus 90% |

**Table 8. The influence of maternal factors on accessing maternal health care services and high risk of maternal deaths.**

| Exposures and Studies | Outcomes | Strength of Difference |
|---|---|---|
| **Maternal health need-associated factors** | | |
| Milkowska-Shibata MA, et al., 2020 [38]<br>Win T, Vapattanawong P and YE P, 2015 [27]<br>Ye *et al.*, 2010 [16] | Demands for maternal healthcare services were higher among women who had a long distance to healthcare facilities than among those who were close to healthcare facilities. | OR = 4.5 (2–10.4) |
| Goland, Hoa and Målqvist, 2012 [39] | Relatively low coverage of reproductive healthcare services was very common among uneducated ethnic minority women. | OR = 6.85 (4.01–11.7) |
| Nguyen PH, *et al.*, 2012 [25]<br>Sychareun *et al.*, 2012 [21] | Having gender-based violence experiences including physical, emotional, and sexual assault was substantially correlated with the unwillingness of taking antenatal care services, induced abortion and recurrent induced abortion. | OR = 1.61 (1.202.16) OR = 2.22 (1.483.32) |

**Table 9. The influence of maternal factors on accessing maternal health care services and high risk of maternal deaths.**

| Exposures and Studies | Outcomes | Strength of Difference |
|---|---|---|
| **Maternal psychological factors** | | |
| Thongmixay S, *et al.*, 2020 [56] | Regarding contraception, non-married women were reluctant to seek contraceptives in public health facilities and because of higher client satisfaction, long-term users were more willing to pay for contraceptive methods: intrauterine device and implant use. | (P<0.001) |
| Fellmeth *et al.*, 2021 [57] | Community, family and husband support were strongly related to maternal depression.<br> Maternal depression was more likely to occur among women having support from their husband, family and environment than those without support. | OR = 2.5 (1.05–5.49) |
| Myo T, *et al.*, 2021 [58] | The relationships between antenatal care services and PPD were noted, -<br>• travel time to health facilities and PPD<br>• frequency of antenatal care and PPD and<br>• post-natal care within 24 hours and PPD | (P<0.0001)<br>(P<0.003)<br>(P<0.05). |

**Maternal psychological factors.** Maternal psychological factors played a significant role in lowering maternal mortality ratios [19] (Table 9). Maternal depression syndromes (MDS), postpartum depression (PPD), and maternal stress (MS) were more common among primigravidae than multigravidae, as were hesitations for childbirth, breastfeeding, and baby care, and primigravidae were afraid to seek maternal healthcare services at public health facilities [18]. It was also reported that the main psychological support factors for promoting maternal health outcomes are respectful maternal care, respect for maternal dignity, and emotional support [51]. Meanwhile, during childbirth, pinching, slapping, and shouting by healthcare providers had a negative impact on maternal psychological conditions [30].

**The status and quality of the maternal healthcare workforce.** Relating to the quality of the maternal healthcare workforce, it was claimed that most maternal deaths came from home deliveries with untrained traditional birth attendances [14, 34]. It was also claimed that the trained birth attendance (TBA) handled pregnancy and childbirth complications in an unscientific and unsafe manner, which could result in maternal deaths [18]. Maternal deaths were found to be affected by the entry of new healthcare providers (e.g., fresher Mid-wives with less expertise on childbirth and revision of maternal complications), the lack of experience during births (e.g., less one year of service), the lack of continuous medical education, and the low quality of basic healthcare providers [28, 33, 58, 59].

Furthermore, it was discovered that the working experiences of health care providers and maternal services were significantly related to the correct decision on maternal emergencies and patient utilization rates (r = 0.301, P0.008) [47, 51].

**Maternal healthcare services use-associated factors.** In CLMV countries, women from some areas were restricted from accessing and using reproductive healthcare facilities, affecting the regularity and quality of antenatal care services and contributing to high maternal mortality ratios. According to one study, the frequency of mothers receiving antenatal care services was very low in hilly, remote, and rural areas although mobile and field maternal healthcare services were provided for higher coverage of antenatal care [15]. Furthermore, many complaints of local pregnant women were reported, including low satisfaction with many supply-side issues, payments for public maternal healthcare services, under-quality services in health facilities, very long waiting times, negligence of patient complaints, and less dutiful responsibilities [18]. Additionally, underdevelopment of healthcare infrastructures and a lack of

medical supplies were identified as major barriers to providing effective care and high-quality services [21, 60].

A study also found that policy and strategic plans for improving maternal mortality ratios were completed only at the highest administrative levels, but were impractical at the implementation level [58]. Another study found that maternal health management information systems, maternal health monitoring and evaluation processes, and health insurance systems were technically and financially deficient [21]. The other studies discussed how, despite the importance of axillary midwives in implementing maternal healthcare services in remote areas, only one training was implemented and no further refresher or formal training was provided [16, 55]. It was argued that the reasons for basic health providers not being less dutiful and responsible for staying in the assigned health facilities full time were due to a lack of any incentives, motivations, educational developments, unsafe conditions and language barriers, a lack of a sufficient supply of essential medicines, and logistics [51, 54, 61].

**Community-associated factors.** The main sources of uncertainty in maternal and reproductive healthcare coverage were a lack of community empowerment and a lack of training for volunteer health workers (VHWs), particularly in hard-to-reach areas [15]. The VHWs could help with the requirements for timely referral (e.g., vehicles, carriers, and manpower), overcoming language barriers, and reducing second delay that is one main cause of maternal death. However, for effective maternal healthcare, community involvement and the availability of VHWs could not overcome some barriers such as long distances and walking time to the nearest medical centers [14, 30, 32, 33]. On the other hand, village leadership and collaboration with healthcare professionals had a greater impact on community-based maternal healthcare services [21]. Furthermore, it was critical for healthcare providers to have mutual respect and effective communication with community leaders in order to improve the quality and demand for maternal healthcare services [51].

## Discussion

In this review, family income was shown to be a strong and significant factor that was substantially related to maternal and reproductive healthcare access and maternal mortality [62]. Pregnant women with low family income were difficult to access quality antenatal care services and were not affordable to actual costs and service delivery charges. In this conclusion, a high proportion of maternal mortality was more significant in pregnant women with low family income. The Nigeria study claimed that the economic development differs among different geographic and income types of the family which in turn affects maternal health care and mortality disproportionately and the high levels of poverty limit access to quality health care and human development index [62]. Actually, the governments of CLMV countries are less ability to propose the initiative programs such as the cost share program for maternal health care services and a pregnancy-peer group fundraising programme. Moreover, the stakeholders in CLMV countries have small numbers of the local health committee and pregnancy and baby love organizations which can pool the necessary funds and support other necessities for poor pregnant women [6].

The degree of maternal education was found to be a significant determinant of maternal mortality in CLMV countries. Education increased the probability that women and their partners had attended antenatal care services, decreased the risk of home deliveries, and improved maternal health outcomes. Illiterate women had a greater likelihood of not obtaining the proposed prenatal and postnatal care and of giving birth with incompetent birth attendants. Similar experiences occurred in Africa and Latin America that women with no education had a 2.7 times greater risk of maternal death than those with one to six years of education, while those

with more than 12 years of education had a two times greater risk [63, 64]. In this issue, the authorities of the Department of Health from CLMV countries insufficiently practice media roles such as (Television, Radio, Facebook, and Channel Programmes) rather than Vinyl, posters, pamphlets, books, journals, magazines and newspapers in maternal health awareness promotion campaigns among illiterate pregnant women. Further, sex education programmes in schools could not continuously be expanded and implemented with the cooperation of the Ministry of Health and the Ministry of Education to distribute maternal and reproductive health messages from students to their families to their communities [6].

Although the child-birth procedure was directly related to pregnant women themselves, this review revealed that the husband and family members played many important roles in maternal healthcare decisions and outcomes. Male involvement was essential for motivating, supporting psychological and material support, and ensuring that mothers lived in safe, happy, and healthy zones. It was also deeply connected with good outcomes for maternal health. These critical experiences occurred in most countries and the study of rural Ghana pointed out that socio-cultural practices of males had impact on maternal mortality. Lack of male involvement will lead to increased maternal mortality and contribute to unwanted pregnancies, pregnancy malnutrition, low use of prenatal and postnatal care facilities, and high rates of illegal abortions [63]. In spite of this importance, the authorities from CLMV countries stills require widely implement intensive public education on the importance of male involvement in the Safe Motherhood Programme and couple counselling practices to promote the roles of the husbands in antenatal care services and during the delivery [6].

Regarding the residence and cultural factors, rural women were more likely than urban women to deliver at home and have birthed with the traditional birth attendance (TBA), showing that geographic constraints had a significant impact on the accessibility of maternal healthcare services and maternal health outcomes. This review found that rural pregnant women had more behaviours of smoking and alcohol drinking and different traditional cultures impacting maternal health. These cultural practices were very similar to the China's hard-to-reach and migration situations. According to the argumentation of China study, pregnant mothers from China's hard-to-reach and migrant areas who had different unacceptable behaviours and cultures were 4.6 times more likely to die from pregnancy and childbirth-related causes, to get illegal deliveries and to reject antenatal and postnatal care provided by the government [65]. Especially, in CLMV countries, there are a smaller number of metropolitan areas that typically have better emergency obstetrics services than rural [6]. Women in rural regions may not be aware of the significance of the conditions that are linked to maternal mortality due to inadequate health facility resources.

Concerning maternal biological factors, there was a correlation between maternal healthcare services preferred and the age of women at marriage, pregnancy, delivery, timing of pregnancies, and the number of children still alive. Early and older ages at marriage were shown to increase the risk of inadequate prenatal care, and older pregnant women were found to be more likely to rely on traditional remedies. Teenage mothers and women who are towards the end of their reproductive years are shown to be at the highest risk of maternal death. Maternal death rates are significantly higher in early marriage and teenage pregnancies (under 16 years old). This may be because young women's pelvic structures are not ready to carry fetuses, which increases the chance of labour obstruction and, in the worst-case scenario, may result in maternal death. In addition, early marriages subject young women to early parenthood and complicated pregnancies that could result in maternal mortality. Because of their physiologic immaturity and general lack of social and economic resources, adolescent females are known to experience several obstetric problems more frequently and with greater severity. These include perinatal mortality, preterm birth, low birth weight, premature birth, anaemia,

malnutrition, cephalopelvic disproportion, vesicovaginal and recto-vaginal fistulae, and pregnancy-induced hypertension. Maternal biological factors were widely discussed in many other studies conducted in developing countries [66–68]. In their revelations, maternal age was significantly associated with maternal mortality, especially older women and adolescents were high-risk groups for maternal mortality. The greatest number of maternal deaths occurred in the age groups of less than 20 years and more than 35 years and among multiparous women. Ageing and a higher risk of pregnancy complications brought on by non-obstetric diseases are contributing factors in older pregnancies (those involving women over 40). Further, the number of pregnancies a woman has affects the risk that she would pass away during pregnancy or childbirth. Women who have their first child (Primigravida, P = 0) or five or more children (Multigravida, P>4) are thought to have an increased risk of dying from maternal causes. Additionally, a close succession of pregnancies and lactation periods worsen the mother's nutritional status because there is insufficient time for the mother to recover from the physiological stresses of the preceding pregnancy before she is subject to the stresses of the next pregnancy. Consequently, it has been discovered that multigravidas had higher rates of maternal death than primigravida [66–68]. To cut the strong association between maternal age and maternal mortality, contraception is available and effective but the expansion of family planning healthcare centers and counselling services are less effective in CLMV countries [6].

Respecting maternal health awareness-associated factors, this review concluded that poor maternal health outcomes were more predominant in mothers who were lack of health awareness of pregnancy-related emergency signs and symptoms, the benefit of antenatal care services, and the effects of unplanned birth. Better maternal health outcomes were strongly correlated with maternal positive attitudes toward ANC services. Women who were more knowledgeable about maternal and reproductive healthcare and related services were 6.5 times more likely to receive antenatal care services than those who were less knowledgeable. The studies in Africa [69] and Nigeria [70]. consistently reported that health awareness of maternal was significantly associated with the maternal health outcome and maternal mortality. The advantage of quality natal care services and effective decisions on the timing of pregnancy, family planning, and birth plan was based on the level of health awareness of pregnant women. It was discovered that literacy level and maternal fatalities were associated, which may be attributed to many effects of knowledge for healthy practice and health service consumption. Therefore, it's crucial to provide knowledge about reproductive health services and promote health to lower maternal mortality.

With reference to community-associated factors, this review could conclude the importance of community empowerment and training and availability of VHWs in maternal healthcare promotion in the rural and remote areas of CLMV countries. A scoping review on the experiences of low- and middle-income countries claimed that only governmental healthcare departments could not provide comprehensive maternal healthcare activities to every mother in different settings and there were many regions of low- and middle-income countries where formal health system could not cover, for which community support and roles of VHWs were valuable [71]. For the poor from rural and remote areas, the communities and HVWs should be active in declining the deaths of their pregnant women and the community-based interventions led by the community leaders and VHWs should be implemented alongside healthcare providers. Understanding the factors that influence maternal mortality at the community level is essential from the standpoint of policy because this is the level at which most policy is implemented. The implementation gap may be evident when the lack of linkages between interventions in health facilities and the communities they serve occurs. Community participation is used as a strategy to understand these local realities, especially in the field of health promotion, and to include local communities in the planning and implementation of health programs

including maternal and child health. Community participation creates an opportunity for community members to share challenges related to maternal and child health and ask for help [72]. Further, the important roles of the local community and VHWs in maternal healthcare improvement that the Ministry of Health in CLMV countries should develop protocols for community-based interventions, home-based intervention packages, community empowerment and engagement, training and incentive programmes for VHWs, and medicines and medicinal supply system.

In respect of maternal health needs-associated factors, this review found that a large proportion of mothers who had no access to professionals' reproductive health care services was one major cause of high maternal mortality in CLMV countries. These constraints were consistently reported in several studies included in this review. China research stated that maternal mortality reduction activities were difficult to achieve due to the main concerns of difficult-to-access areas which in turn relate to the development of other barriers such as language and cultural barriers, transportation barriers, socio-economic barriers, and negative perceptions toward maternal healthcare [73]. Similarly, a Ghana study supported these findings that many socio-cultural barriers to maternal and reproductive health care were the consequences of inaccessible refrainment [74]. From this issue, this review recommended three options. The first one was that the governments of the CLMV countries should allocate more budget for upgrading and expanding rural healthcare facilities and maternal healthcare packages and prioritizing the development of roads and transportation infrastructures to medical centres to be approachable to maternal and reproductive healthcare services within minimal hours. The second one was that the Ministry of Health in CLMV countries should develop and implement more collaborative actions with local leaders, local health authorities and other non-profit organizations focusing on different supplies of maternal health needs. The third one was that the Department of Health and Education in CLMV countries should practice local-tailored approaches such as opening local birth centres, establishing community midwife programmes, and implementing peer education programmes for women.

Another important revelation of this review on high maternal mortality in CLMV countries was the inequalities in the types of maternal healthcare services and the quality of healthcare providers. In CLMV countries, approximately 2 in 10 mothers came from hard-to-touchable areas and they had not received maternal health-related services such as examination of pregnancy, HIV and Syphilis counselling and testing, deworming, iron and Vitamin B1 and A supplementation, TT immunization, contraceptive and nutritional counselling, and care of pregnancy- and childbirth-related complications. These experiences in CLMV countries were very similar to the events in India's hilly and valley regions. An Indian study on inequalities in maternal healthcare access between the hilly and valley areas revealed that antenatal care visits, institutional deliveries, and other supporting services for maternal health were very limited, leading to high maternal mortality in these areas [75]. From this conclusion, this review recommended that the health authorities from the Department of Health from the regions in CLMV countries, where maternal mortality was high, should provide special efforts such as collecting maternal base-line data and estimating the possible target of pregnancy, estimating and supplying maternal health-related drugs and equipment, training the existing local traditional healers, recruiting maternal healthcare promotors, supervising and evaluating the quality of maternal healthcare services and conducting scientific research. Additionally, local government authorities should look for more effective approaches and strategies in bringing changes in maternal healthcare-related issues including vocational training for women, provision of opportunities for reading and focus group discussions, and other options impacting maternal education, economics and psychology.

Despite efforts to speed up comprehensive maternal healthcare and reduce maternal mortality, the CLMV countries faced a serious shortage of maternal healthcare workforce, especially in rural settings. Regarding the maternal healthcare workforce, this review concluded that most of the maternal mortality was due to a shortage of healthcare workforce in the CLMV countries. These critical challenges occurred in many countries in different forms including a shortage of qualified healthcare providers, a shortage of healthcare workforce, unequal distribution of skilled healthcare providers, and limited access to skilled healthcare providers [76]. From these issues, this review recommended that the recruitment policy of healthcare providers should be based on residence and locality of assigned staff, the entrance of medicinal universities should be based on local manpower needs rather than academic qualification, and healthcare providers operating in remote areas should have more chance of career development and their pay scales should be double or triple and other additional incentives should be more in healthcare providers in hard-to-reach areas. Besides, this review concluded that many maternal deaths in CLMV countries arose from the deliveries of unskilled traditional birth attendance, which did not mean due to unskilled birth attendance and their root causes were poor knowledge and unavailability of delivery equipment. An Indian study on the performance of Axillary Nurse Midwives (ANM) claimed that training and recruiting local ANM is an effective intervention for overcoming the shortage of the maternal healthcare workforce [77]. Accordingly, this review also recommended that health authorities in CMLV countries should conduct training and recruitment for a sufficient number of local axillary nurse midwives. The scope of work of axillary nurse midwives should be expanded and their necessary skills should cover abortion care, delivery management, the administration of misoprostol, oxytocin, and intravenous fluids, bimanual uterine compression for postpartum haemorrhage, management of minor lacerations, infection control and management, management for maternal emergencies, management of puerperium sepsis, administration of magnesium sulphate, TT immunization and well-preparing referral techniques. Moreover, the training should be ongoing, supervised and refreshed.

In relation to the associated factors of maternal healthcare service utilization, this review reported that low utilization rates of natal care services among women from rural or remote areas were due to supply-side constraints. Many studies included in this review revealed that different supply-side constraints that occurred in CLMV countries included increased absenteeism, inabilities to retain competent healthcare professionals, inconsistent assignment of the healthcare workforce, weak technical guidance and poor human resource management. Besides, this review also reported that mobile and outreach maternal healthcare services were very few or lacked in the difficult-to-reach areas in CLMV countries, consequently, service utilization rates were low. These similar experiences were found in the maternal and reproductive health programmes in India [78], Western Kenya [79], Northern Ghana [80] and Malawi [81]. The studies from these countries consistently reported that the distance and time, socioeconomic factors of community and the availability of staff, medicines and medicinal equipment were barriers to comprehensive maternal healthcare between provider sites and service user sites [78–81]. To reduce supply-side constraints affecting maternal healthcare service utilization in CLMV countries, this review recommended that health authorities from the Department of Human Resource Management should estimate and balance the minimal adequacy of health staff, and the Department of Education should provide appropriate, sufficient and follow-up training on all available staff, the Department of Health should balance staff's availability and duties with their likelihood needs and the Regional and District Health Departments should develop strategic tour plans of maternal healthcare packages for the remote areas and implement the plans as schedules and with the highest quality.

This review concluded that the maternal healthcare utilization rates were low due to the dissatisfaction of mothers with service provision-related issues such as out-of-pocket expenses, few servicing hours, violation of patient privacy, poor and unclean infrastructures, prolonged waiting times, and lack of adequate and private space for pregnancy- and childbirth-related services. These challenges were consistently reported outsides of CLMV countries [78–81]. From this point of view, this review recommended that the National Health Implementation Systems from CLMV countries should allocate timely pay for purchasing and repairing basic infrastructures and essential inputs for maternal healthcare services, and expand and assign healthcare counselling sections and counsellors. To reduce the high congestion of maternal cases and respect patient privacy, comprehensive maternal health facilities, laboratory facilities and other one-stop service facilities should be provided alongside by either constructing more maternal health compounds or upgrading the existing ones. Besides, maternal health-related policy and strategic plans were on paper and not effective for improving maternal mortality outcomes. As recommended by other studies on their findings, every strategic plan developed for maternal health needs to be community-based practices and actually at the community levels for allowing mothers to be healthy and safe motherhood and policy-makers themselves to achieve the sustainable development goals [78–81].

With regard to maternal psychological factors, this review summarized that maternal depression syndrome, maternal emotion and maternal stress were more likely to occur among primigravida, and non-married women were unwilling to disclose their contraception status. A recent Myanmar study reported that about 22% of mothers with first Gravida and more than 38% of mothers with the second gravida had experienced postpartum depression syndrome [57]. The studies claimed that depression during pregnancy led to poor prenatal and postnatal care, delays in seeking care for pregnancy-related complications and potential for maternal deaths [82, 83]. The Myanmar study also showed the reasons for developing postpartum depression that the primigravidae were fear of giving childbirth, delivery complications, negligence of birth attendance, interruption of education and regular income, negligence of their spouses and lack of experience in baby care [57]. To prevent maternal depression-related deaths, this review recommended that antenatal and postnatal counselling services and enquiring the previous psychiatric history should be added to the existing maternal and reproductive healthcare services in CLMV countries. The husbands and family members should be targeted in the maternal health awareness promotion programmes. The Department of Health in CLMV countries should develop the protocols for maternal psychiatric management and train the basic health staff on how to investigate and manage maternal psychological conditions.

## Conclusion

Maternal mortality is still sensitive and increasing in low-income countries and globally critical issue for public health. Since high maternal mortality affects the quality of community development and responsiveness of the healthcare system, reducing maternal mortality ratios is a health priority in every country especially low- and middle-income countries including CLMV countries. Maternal mortality is associated with different factors: health system-related factors, healthcare programme-related factors, healthcare providers-related factors, community-associated factors and maternal associated factors. Many of these factors are revisable and need to be tackled urgently, if so, this review will be capable of identifying several issues affecting maternal mortality and summarizing these factors useful for the development of maternal healthcare-related strategies and policies. This systematic review mainly concluded that the underlying factors of a higher overall maternal mortality in CLMV countries are due to different in the healthcare systems, unequal access to resources, inadequate medical care,

unaffordable healthcare, lower socio-economic status and unhealthy behaviours of women, less promotion of maternal health awareness, maternal psychological disorders and poor involvement of husbands and families in maternal healthcare services. Overall, the results of this evaluation will advance knowledge about maternal healthcare and mortality and provide a valuable summary to policymakers and faculty members for developing policies and strategies promoting high-quality maternal health care.

## Limitations of study

This review could pick up acceptable percentages of relevant studies from each CLMV country and therefore the findings of the review might raise the geographical representativeness, but could not reflect the importance of other regions. This review excluded some forms of publication; books, government reports, conferences, meeting proceedings and purchased papers. Thus, this review might miss some significant findings disseminated in these publication forms.

## Supporting information

**S1 Checklist. PRISMA 2020 checklist.**
(DOCX)

**S1 Dataset.**
(XLSX)

## Acknowledgments

The corresponding author, Dr Pyae Phyo Win sincerely and submissively gives special thanks to Professor Dr Hla Hla Win (Supervisor), Mr Thein Hlaing (Co-reviewer), the lecturers: Professor Dr Kyaw Myo Tun, Professor Dr Pa Pa Soe, Professor Dr Kay Thi Lwin, Dr Than Tun Sein, Dr Mon Mon Aung, Dr Tin Tin Aye, Dr Aung Phone Zaw, and Dr, Ye Yint, the course administrators: Professor Cho Mar Lwin, Dr Thinn Yu Aung, and Ko Sithu for their brilliant academic knowledge, erudite support, guidance, and encouragement and facilitations.

## Author Contributions

**Conceptualization:** Pyae Phyo Win, Thein Hlaing, Hla Hla Win.

**Data curation:** Pyae Phyo Win, Thein Hlaing, Hla Hla Win.

**Formal analysis:** Pyae Phyo Win, Thein Hlaing.

**Funding acquisition:** Pyae Phyo Win.

**Investigation:** Pyae Phyo Win, Thein Hlaing.

**Methodology:** Pyae Phyo Win, Thein Hlaing.

**Project administration:** Pyae Phyo Win, Thein Hlaing, Hla Hla Win.

**Resources:** Pyae Phyo Win, Thein Hlaing.

**Software:** Pyae Phyo Win, Thein Hlaing.

**Supervision:** Pyae Phyo Win, Thein Hlaing, Hla Hla Win.

**Validation:** Pyae Phyo Win, Thein Hlaing, Hla Hla Win.

**Visualization:** Pyae Phyo Win, Thein Hlaing, Hla Hla Win.

**Writing – original draft:** Pyae Phyo Win.

**Writing – review & editing:** Pyae Phyo Win, Thein Hlaing, Hla Hla Win.

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
