## [Decision Letter · Decision Letter 0]

13 Jul 2023

PONE-D-22-35646Factors influencing maternal death in Cambodia, Laos, Myanmar, and Vietnam (CLMV) countries:

A systematic reviewPLOS ONE

Dear Dr. win,

Thank you for submitting your manuscript to PLOS ONE. After careful consideration, we feel that it has merit but does not fully meet PLOS ONE’s publication criteria as it currently stands. Therefore, we invite you to submit a revised version of the manuscript that addresses the points raised during the review process.

We look forward to receiving your revised manuscript.

Kind regards,

Ammal Mokhtar Metwally, Ph.D (MD)

Academic Editor

PLOS ONE

Journal Requirements:

Additional Editor Comments:

The manuscript is interested meanwhile, the reviewers have raised a number of points which we believe would improve the manuscript and may allow a revised version to be published in PLOS one.

Reviewers' comments:

Reviewer's Responses to Questions

**Comments to the Author**

1. Is the manuscript technically sound, and do the data support the conclusions?

Reviewer #1: Yes

Reviewer #2: Partly

2. Has the statistical analysis been performed appropriately and rigorously? 

Reviewer #1: Yes

Reviewer #2: N/A

3. Have the authors made all data underlying the findings in their manuscript fully available?

Reviewer #1: Yes

Reviewer #2: Yes

4. Is the manuscript presented in an intelligible fashion and written in standard English?

Reviewer #1: No

Reviewer #2: Yes

5. Review Comments to the Author

Reviewer #1: The authors well prepared review, but I can say the way they presented the study has exagerated text, a lot of repetations and unimportnat texts is observed in every section of the study, for example the aim of the study was repeated more than twice in different places-at the first paragraph of the methods for instance. Inclussion and exclussion chart was not observed as it would enlight that process more easily. As we are reviwing secondry data, it is better to present the study population as such, when you mention study population it sounds like primary study, the male population that you mentioned the inclussion creteria was not observed in the analysis part.

Data analysis, you mentioned a lot of information about the existing qualitative data analysis method which is not relevant, and increasing unncessary text (you mentioned content analysis, narrative analysis, discourse analysis, framwork analysis and grounded theory).

To flow the dat more easily, if the tables are long it is easier to use appendix instead of putting the long tables within the text.

Lastely the discussion was good but again very much text and repeated statements.

Reviewer #2: please see the comments and modify accordingly. All the best

The authors need to arrange introduction section. Rewrite some sentences so, the readers will understand what you wanted to convey from messages. The authors should make sure that missing references are written within the introduction section. They also need to make sure to clearly mention the significant of the systematic review and the aim of it. The authors should indicate what they wanted the readers to understand. The authors should indicate the significance of the systematic review and the overall aim of this systematic review. See the remaining comments

6. PLOS authors have the option to publish the peer review history of their article (what does this mean?). If published, this will include your full peer review and any attached files.

Reviewer #1: No

Reviewer #2: **Yes: **Zalikha Al-Marzouqi

---

## [Author Response · Author response to Decision Letter 0]

27 Aug 2023

I would like to express my gratitude for the prompt and thoughtful review of my manuscript submitted to PLOS ONE. Your valuable feedback is greatly appreciated, and I am eager to address the points raised during the review process to improve the quality and alignment of the manuscript with PLOS ONE's publication criteria.

I have taken careful note of the specific areas that need attention, and I am committed to revising the manuscript accordingly. Your guidance is instrumental in enhancing the overall clarity and impact of the research.

---

## [Decision Letter · Decision Letter 1]

2 Oct 2023

PONE-D-22-35646R1Factors influencing maternal death in Cambodia, Laos, Myanmar, and Vietnam (CLMV) countries:

A systematic reviewPLOS ONE

Dear Dr. win,

Thank you for submitting your manuscript to PLOS ONE. After careful consideration, we feel that it has merit but does not fully meet PLOS ONE’s publication criteria as it currently stands. Therefore, we invite you to submit a revised version of the manuscript that addresses the points raised during the review process.

We look forward to receiving your revised manuscript.

Kind regards,

Ammal Mokhtar Metwally, Ph.D (MD)

Academic Editor

PLOS ONE

Journal Requirements:

Additional Editor Comments:

Thank you for addressing all reviewers' comments. We believe your manuscript is ready for publication after considering the following comments: the inclusion of the inclusion and exclusion criteria chart and deleting the repetitive parts of the discussion

Reviewers' comments:

Reviewer's Responses to Questions

**Comments to the Author**

1. If the authors have adequately addressed your comments raised in a previous round of review and you feel that this manuscript is now acceptable for publication, you may indicate that here to bypass the “Comments to the Author” section, enter your conflict of interest statement in the “Confidential to Editor” section, and submit your "Accept" recommendation.

Reviewer #1: (No Response)

Reviewer #2: All comments have been addressed

2. Is the manuscript technically sound, and do the data support the conclusions?

Reviewer #1: Yes

Reviewer #2: Yes

3. Has the statistical analysis been performed appropriately and rigorously? 

Reviewer #1: Yes

Reviewer #2: N/A

4. Have the authors made all data underlying the findings in their manuscript fully available?

Reviewer #1: Yes

Reviewer #2: Yes

5. Is the manuscript presented in an intelligible fashion and written in standard English?

Reviewer #1: Yes

Reviewer #2: Yes

6. Review Comments to the Author

Reviewer #1: It seems that the author har addressed the comments, it would have been good to include the inclussion and exlcussion creteria chart, the discussion part is still extenssive and repetetive, otherwise it is OK.

Reviewer #2: The authors addressed all comments for this review. Well done, wishing the authors all the best in publication.

7. PLOS authors have the option to publish the peer review history of their article (what does this mean?). If published, this will include your full peer review and any attached files.

Reviewer #1: No

Reviewer #2: **Yes: **Dr. Zalikha Al-Marzouqi

---

## [Author Response · Author response to Decision Letter 1]

5 Oct 2023

Thanks for kind of your information.

---

## [Editor Report · Decision Letter 2]

8 Oct 2023

Factors influencing maternal death in Cambodia, Laos, Myanmar, and Vietnam (CLMV) countries:

A systematic review

PONE-D-22-35646R2

Dear Dr. win,

We’re pleased to inform you that your manuscript has been judged scientifically suitable for publication and will be formally accepted for publication once it meets all outstanding technical requirements.

Kind regards,

Ammal Mokhtar Metwally, Ph.D (MD)

Academic Editor

PLOS ONE

Additional Editor Comments (optional):

Thank you for addressing all reviewers' comments. We believe your manuscript is ready for publication.
---

## [Editor Report · Acceptance letter]

25 Oct 2023

PONE-D-22-35646R2 

Factors Influencing Maternal Death in Cambodia, Laos, Myanmar, and Vietnam Countries: A Systematic Review  

Dear Dr. Win:

I'm pleased to inform you that your manuscript has been deemed suitable for publication in PLOS ONE. Congratulations! Your manuscript is now with our production department. 

Kind regards, 

on behalf of

Professor Ammal Mokhtar Metwally 

Academic Editor

PLOS ONE